# Physical and Physiological Responses during the Stop-Ball Rule During Small-Sided Games in Soccer Players

**DOI:** 10.3390/sports7050117

**Published:** 2019-05-17

**Authors:** Jamel Halouani, Kais Ghattasi, Mohamed Amine Bouzid, Thomas Rosemann, Pantelis T. Nikolaidis, Hamdi Chtourou, Beat Knechtle

**Affiliations:** 1Research Unit, Education, Motricity, Sport and Health, UR15JS01, High Institute of Sport and Physical Education, University of Sfax, Sfax 3000, Tunisia; jamelhal@yahoo.fr (J.H.); kaisgatassi@yahoo.fr (K.G.); bouzid.mohamed-amine@hotmail.fr (M.A.B.); 2Institute of Primary Care, University of Zurich, 8091 Zurich, Switzerland; thomas.rosemann@usz.ch; 3Exercise Physiology Laboratory, 18450 Nikaia, Greece; pademil@hotmail.com; 4Institut Supérieur du Sport et de l’éducation physique de Sfax, Université de Sfax, Sfax 3000, Tunisie; h_chtourou@yahoo.fr; 5Activité Physique, Sport et Santé, UR18JS01, Observatoire National du Sport, Tunis 1003, Tunisie; 6Medbase St. Gallen Am Vadianplatz, 9001 St. Gallen, Switzerland

**Keywords:** soccer, small-sided games, physical performance

## Abstract

Small-sided games (SSGs) are a recommended training method for significant performance enhancement, and training efficiency. The stop-ball (SSG-SB) effects on physical responses (e.g., acceleration, deceleration, sprints, total distance, and indicator of workload) have not been investigated yet. This study aimed to investigate the effects of the SSG-SB compared to the small-goals SSG (SSG-SG) on physical and heart rate (HR) responses at high intensity (total distance (>18 km/h)), sprints (>18 km/h), and acceleration and deceleration (>3 m/s²) during a 4 vs. 4 SSG format in youth professional soccer players. Sixteen male elite young soccer players (mean ± SD body height, 176.5 ± 6.3 cm; age, 18.3 ± 0.7 years; body weight, 73.4 ± 7.2 kg) performed two forms of SSGs, i.e., SSG-SB or SSG-SG, for 4 × 4 min with a recovery of 2 min between sets. Data were compared using the *t*-test. The SSG-SB induced a significantly higher mean HR (180.0 ± 2.0 vs. 173.0 ± 3.0 beats per minute; *p* < 0.05) compared to the SSG-SG. Likewise, the SSG-SB was significantly higher compared to the SSG-SG for total distance (2580 ± 220.3 vs. 2230 ± 210 m; *p* < 0.001), player load (98.07 ± 12.5 vs. 89.4 ± 10.5; *p* < 0.05), sprint distance (7.9 ± 2.3 vs. 5.2 ± 2.0 m; *p* < 0.05), acceleration (15.6 ± 2.75 vs. 12.5 ± 1.75; *p* < 0.05), and deceleration (17.3 ± 3.20 vs. 14.4 ± 2.55; *p* < 0.05). However, no significant difference was found between the SSG-SG and the SSG-SB for maximal velocity, power, and sprints duration. This study provides new information about the effectiveness of the SSG-SB as a training stimulus for soccer.

## 1. Introduction

Soccer is a highly physically demanding sport, where both the aerobic and anaerobic pathways are considerably taxed [1]. During a match, players typically cover a distance of 10–13 km, performing 150–250 intense actions (e.g., accelerations/decelerations, changes of direction) interspersed with short recovery periods [2]. Therefore, physical conditioning for professional soccer players is of utmost importance.

Small-sided games (SSGs) training combines physical, technical, and tactical components, and has been recommended for performance enhancements [3]. Moreover, the previously reported beneficial effects of this training method justify its wide utilization within professional-level soccer as aerobic capacity training [4,5,6,7]. Indeed, changing tactical and technical constraints [8], i.e., bout duration [9], player numbers [10,11], pitch size [12,13], and types of scoring [14], modifies the physiological responses to SSGs.

Concerning the types of scoring in SSGs, Halouani et al. [14,15,16] found that the stop-ball SSG (SSG-SB: stop the ball in a 1-m wide zone behind the end line) induced higher heart rate (HR) values than the small-goal SSG (SSG-SG: score a goal in small goals placed at the center of the end line) in three pitch sizes (i.e., 10 × 15, 15 × 20, and 20 × 25 m) and for three game formats (i.e., 2 vs. 2; 3 vs. 3; and 4 vs. 4 players). Therefore, coaches could utilize the SSG-SB as an alternative for increasing metabolic and cardiovascular demands in young soccer players. The players participating in these studies were young (average age 13.6 years) and playing in an amateur league. Therefore, the confirmation of these observations with elite soccer players seems to be of importance for coaches, athletes and scientists. Moreover, although the studies by Halouani et al. [14,15,16] described the influence of two forms of SSG on physiological responses, the physical response to the SSG-SB has not yet been investigated.

The use of the portable global positioning system (GPS) has previously been reported for recording time-motion characteristics during a game, including physical parameters for all types of soccer training. Therefore, it is now possible to examine the influence of two forms of SSG (SSG-SB and SSG-SG) on both physical parameters and HR responses.

Although the manipulation of scoring type [14] has been reported to affect the physiological responses to SSGs, no previous study has compared the physical response to the SSG-SB and the SSG-SG. In view of the importance and the effectiveness of these two training methods (i.e., SSG-SB and SSG-SG) to increase intensity in SSGs, this study aimed to compare the physical and physiological responses to the SSG-SB and the SSG-SG in elite young soccer players. We can conclude that this comparison could be useful for players and coaches during SSG training. In view of the previous published studies, it was hypothesized that the SSG-SB would elicit higher physiological and physical demands than the SSG-SG.

## 2. Materials and Methods

### 2.1. Subjects

Sixteen elite young male soccer players voluntarily participated in this investigation (mean ± SD age, 18.3 ± 0.7 years; body height, 176.5 ± 6.3 cm; body weight, 73.4 ± 7.2 kg). They were all members of the same team competing in the Tunisian elite young league, and they had a mean of eight years of experience in playing soccer. Written informed consent was obtained after all players were informed about the protocol of the study. The study protocol was approved by the Sfax University ethics committee (CPP-02/18).

### 2.2. Experimental Approach to the Problem

In this study, physical and HR responses to the SSG-SB and to the SSG-SG have been compared using the same number of players (i.e., 4 vs. 4), pitch dimensions (i.e., 20 × 25 m), playing time, and recovery (i.e., 4 × 4 min with 2 min recovery between sets). Although different formats of SSGs were usually performed by the participants in their training programs (i.e., including the SSG-SB and the SSG-SG), they underwent a supplementary familiarization session before the start of the experimental phase, with an explanation of the required tasks and the progress of each test session.

### 2.3. Experimental Procedures

The SSG-SB and the SSG-SG were conducted on different days (i.e., one for the SSG-SB and one for the SSG-SG) and at the same time of day (16h00 to 18h00) [17]. A point was scored when the player stopped the ball behind the bottom line of playing field (i.e., a zone of 20 × 1 m) during the SSG-SB (Figure 1). Stopping the ball means finding a way of entering the “goal zone” with the ball and stopping the ball under the sole of one foot. A ball transiting into the zone was not sufficient to obtain a goal. However, for the SSG-SG, a point was scored when the player scored in a small goal (i.e., 1 m width and 0.5 m height) (Figure 2). All players were instructed to move and not to defend by staying nearby the goal. No goalkeepers were used during the SSG-SG. Both types of SSGs were performed without a limitation on the number of ball contacts. Verbal encouragement was continuously offered during the SSG-SB and the SSG-SG [7]. To minimize lost time, extra balls were placed around the playing surface. A standard ~15-min warm-up was performed before both the SSG-SB and the SSG-SG (i.e., running and dynamic stretching followed by a ball-specific warm-up).

Data (i.e., acceleration, deceleration, total distance, sprints, and indicator of workload) were collected using wearable GPS tracker devices (Playertek, Catapult Innovations, sampling frequency = 10 Hz, Melbourne, Australia) that provide a high degree of locational accuracy with minimal battery impact, and can track the player with a high resolution across the field to generate comprehensive movement maps. The GPS device was activated 15 min before kick-off. After recording the two types of SSGs, all of the data gathered by the Playertek Pod was viewed and interpreted on the Playertek app, which is provided free of charge with the earth Playertek platform. The analysis of the GPS data was based on four zones of speed [9,18]: 0–6.9 km/h (walking), 7.0–12.9 km/h (low-intensity running), 13.0–17.9 km/h (moderate-intensity running), and >18 km/h (high-intensity running). Similarly, sprints (i.e., duration and distance) were recorded based on these four speed zones. Acceleration distances were recorded using the category (2–3 m/s^2^ and >3 m/s^2^) and the type (i.e., acceleration or deceleration) [19]. Moreover, global load indicators were also measured: maximal velocity, power, and player load. The Polar Team Sports System (Polar Electro Oy, Kempele, Finland) was used to record the HR responses at an interval of 5-Hz during the SSG-SB and the SSG-SG and the HRmax (maximum heart rate) was calculated based on the age (HRmax = 220 - age). HR mean values were expressed in absolute values (beats per minute (bpm)) and were utilized to determine the utilized percentage of HRmax (%HRmax) during both SSG formats.

### 2.4. Statistical Analysis

Data were analyzed using the software package STATISTICA (StatSoft^®^, Maisons-Alfort, France) and were reported as mean and SD. The normality of the data was confirmed using the Shapiro–Wilk *W*-test; and the paired Student’s *t*-test was used to compare the SSG-SG to the SSG-SB. Effect size was calculated using Cohen’s d and classified as no effect (d < 0.41), minimum (0.41 < d < 1.15), moderate (1.15 < d < 2.70), and strong effect (d > 2.70) [20]. Intraclass correlation (ICC) and the coefficient of variation (CV) were calculated in order to determine the reliability of the measured parameters. A significant difference was accepted at *p* < 0.05.

## 3. Results

Test-retest reliability for HR, total distance covered, player load, sprint distance, and acceleration as measured by the CV and ICC were 7.6% (ICC = 0.88 95% CI: 0.60–0.96), 8.7% (ICC = 0.73 95% CI: 0.1–0.86), 9.5% (ICC = 0.78 95% CI: 0.2–0.91), 10% (ICC = 0.75 95% CI: 0.5–0.88), and 9.6% (ICC = 0.80 95% CI: 0.3–1), respectively.

The statistical analysis revealed that HR and %HRmax were higher during the SSG-SB compared to the SSG-SG (Table 1). In addition, the statistical analysis revealed that total distance, walking distance, low-intensity running distance, moderate-intensity running distance, and high-intensity running distance were significantly higher during the SSG-SB compared to the SSG-SG (Table 1). Likewise, player load, sprints distance, acceleration and deceleration were significantly higher during the SSG-SB compared to the SSG-SG (Table 1).

## 4. Discussion

Examining the physical and physiological (HR and %HRmax) responses to the SSG-SB and the SSG-SG in elite young soccer players, the main findings were that the SSG-SB induced higher physical and HR values than the SSG-SG.

The HR response of players is a useful method to regulate the intensity of training [21]. The results of this study showed that the HR was significantly higher during the SSG-SB than the SSG-SG for the 4 vs. 4 format. Furthermore, the SSG-SB induced higher intensities (HR values and HRmax) compared with the SSG-SG (180 vs. 173 bpm; 89.68 vs. 85.83%, respectively). In this context, only the studies of Halouani et al. [15,16] have compared the physiological responses to the SSG-SB versus the SSG-SG on amateur young soccer players using three formats of players’ number (i.e., 2 vs. 2, 3 vs. 3, and 4 vs. 4).

The present study confirmed the results of Halouani et al. [15], which reported a higher SSG intensity (i.e., higher HR values and %HRmax) during the SSG-SB compared to the SSG-SG. These findings could be explained by: (i) the larger area that the defensive and the attacking players must cover in the SSG-SB, (ii) the motivation factor (players are motivated by this new form of scoring), and (iii) less technical abilities during the SSG-SB format (i.e., the SSG-SB required less technical abilities than the SSG-SG, as the scoring zone is large). In addition, we speculate that, with this new form of scoring, there is less tactics and more running. Despite the differences between the present study and the studies of Halouani et al. [15,16] regarding age (18.3 vs. 13 years) and the player category (elite vs. amateur), the present study is in agreement with the results of Halouani et al. [15,16] during the 4 vs. 4 SSG, and confirmed the higher SSG intensities (HR) observed during the SSG-SB than the SSG-SG. Moreover, Jones and Drust [5] have compared the physiological responses of two forms of SSGs (4 vs. 4 and 8 vs. 8 with goalkeepers). The HR values of the 4 vs. 4 in the study of Jones and Drust [5] was similar to the 4 vs. 4 SG-SSG observed in the present study (i.e., 175 vs. 173 bpm). This result could be explained by the fact that the inclusion of goalkeepers reduced the intensity of the SSGs compared to playing and conserving the ball without scoring. Also, it is possible that some players were more motivated than others by the presence of goalkeepers, explaining such a difference. This result suggests that the stop-ball rule is related to a new form of scoring a goal and may influence the player’s motivation to increase or maintain exercise intensity and therefore enhance the player’s physiological response to SSGs.

This study is the first to describe the physical parameters associated with two forms of SSGs in elite young soccer players. There is a very important relationship in soccer SSGs between the external loads and the physiological responses. Nowadays, different technological devices, such as GPS devices, offer a highly practical way of monitoring players’ movements during training and SSGs [22]. In this study, we can show the difference on the external load of SSGs between two game rules (i.e., stop-ball vs. small-goal rules) using the 4 vs. 4 format.

Distances covered by players at various speeds were measured in addition to the physiological responses, to determine the exercise intensity during SSGs [23]. In the present study, it was observed that both the total distance and the total distance covered in high-speed running was greater during the SSG-SB compared to the SSG-SG. Therefore, compared to the SSG-SG, in the SSG-SB it is possible that: (i) players do not need to shoot the ball, (ii) the ball is in play for a longer time, and (iii) the encouragement factor can help players to cover longer distances at higher running speeds and total distances. In the same context, the speed of play increases in the SSG-SB due to pressure from opponents that can induce high-speed movements and can enhance the amount of running performed by players. Moreover, the number of accelerations and decelerations (>3 m·s^−^²), which is an essential element of soccer, was higher during the SSG-SB than the SSG-SG (acceleration: 15.6 vs. 12.5 m·s^−^² and deceleration: 17.3 vs. 14.4 m/s², respectively). In this context, Rebelo et al. [19] showed a greater number of accelerations and decelerations performed in the 4 vs. 4 than in the 8 vs. 8 format, which could reflect a higher metabolic and mechanical loading of the neuromuscular system in the smaller format. Thus, the 4 vs. 4 SSGs format induces high intensity, in relation to repetitions and fatigue development in muscle, in comparison with other SSG formats. This information is very important in weekly training for soccer players. As eccentric exercises impose a high mechanical strain [24], requiring long recovery times, coaches should not schedule SSGs with a low number of players too close to official matches in order to prevent unnecessary fatigue. Furthermore, previous studies showed that rule modifications could influence physical responses during SSGs. For example, Hill-Haas et al. [18] and Köklü et al. [25] examined the physical responses of youth players in three different SSG formats (i.e., 2-a-side, 4-a-side, and 6-a-side) with a constant pitch area. Hill-Haas et al. [18] and Köklü et al. [25] also reported that SSG rule modifications can affect physical responses. In this study, results show that the new rule used (stop-ball) influenced external load in the SSGs, with higher physical responses recorded during the SSG-SB than the SSG-SG.

Altogether, the findings of this study suggest that the SSG-SB played with the 4 vs. 4 format on a constant pitch provides higher intensities (HR and physical responses) than the SSG-SG. Thus, the use of this form of SSG throughout the course of the training week is something that should be noted in order to maximize the preparation of players and expose players to varying physical and physiological demands within a more controlled environment. The fact that this SSG form (i.e., SSG-SB; 4 vs. 4; constant pitch) has shown high intensities in a training environment may only serve to add to the confidence of knowing the specific physical and physiological requirements of this game when organizing and planning training. Then, to evaluate the intensity of SSGs we must associate external and internal loads.

This study has a number of strengths and limitations that should be acknowledged. A limitation of this study was that the physiological responses to an SSG have been shown to vary by its specific characteristics (e.g., number of players, duration, pitch area, and rules) [19]. Also, the percentage of HRmax, which is estimated based on age, could be influenced by individual variability. Moreover, the results of this study could be partially due to the environment unique to the team and the coaching staff. Thus, caution would be needed when modifying or introducing rules in the SSG formats to adjust them to the competition demands. In addition, a study strength was its novelty, as it was the first study to examine the stop-ball rule in a 4 vs. 4 SSG format with elite soccer players and its effect on HR and physical parameters. Considering the wide use of SSGs as a training tool, these findings would be of great practical relevance for coaches and fitness trainers working with soccer players to manage exercise intensity using the stop-ball rule.

## 5. Conclusions

In the present study, we compared the effects of two forms of SSG (stop-ball vs. small-goals) in elite young soccer players on HR and on the physical responses, e.g., acceleration, deceleration, sprints, total distance, and indicator of workload. The SSG-SB appeared to be more demanding and more intense than the SSG-SG on HR and physical parameters. The main practical application, for coaches and strength-conditioning professionals, to be drawn from this study is that changes in game rules (SSG-SB and SSG-SG) affect the players’ physiological and physical demands. This may enable sports science and conditioning staff to optimally prepare players physically, thus increasing the efficiency of their training sessions and weekly schedule. Similarly, this research is useful for coaches, and the findings should be taken into account when designing SSG training of elite soccer players. Changes in any rule during SSGs should be introduced thoughtfully, because the effect on physiological and physical responses appears to be influenced by the motor competence of the players involved and by the skill and fluidity with which they are able to perform the drill.

## Figures and Tables

**Figure 1 sports-07-00117-f001:**
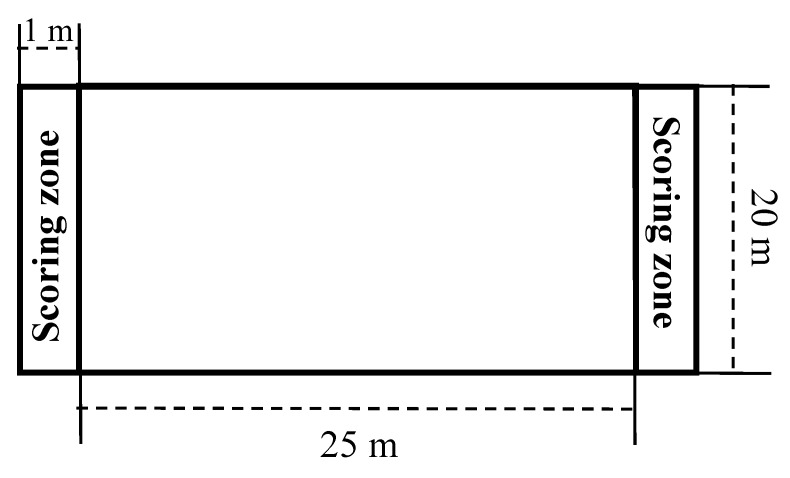
Small-sided game with Stop-ball.

**Figure 2 sports-07-00117-f002:**
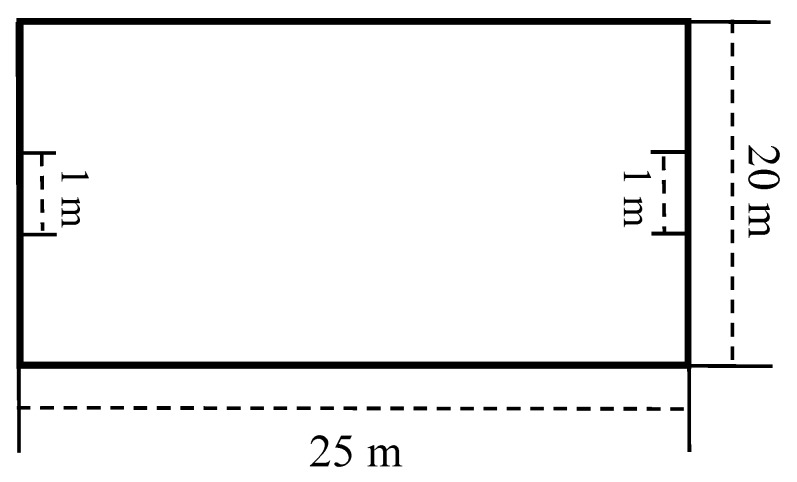
Small-sided game with small-goals.

**Table 1 sports-07-00117-t001:** Mean heart rate (HR) values, percentage of maximum heart rate (%HRmax), total distance, walking distance, low-intensity running distance, moderate-intensity running distance and high-intensity running distance, player load, sprints distance, and acceleration and deceleration recorded during a stop-ball small-sided game (SSG-SB) and a small-goal small-sided game (SSG-SG).

Heart Rate	SSG-SB	SSG-SG	t	D	*p*	95% CI
**HR and %HRmax**						
HR (bpm)	180 ± 2.75 *	173 ± 3.02	2.58	2.42	<0.05	3.90 to 10.10
%HRmax	89.68 ± 3.71 *	85.83 ± 1.37	2.58	1.37	<0.05	0.85 to 6.85
**Total Distance (Four Zones of Speed)**						
Total distance during the 4 × 4 min (m)	2580 ± 220.3 *	2230 ± 210	10.86	1.62	<0.001	69.21 to 530.79
Walking during the 4 × 4 min (m)	1120.8 ± 100.2 *	1005.7 ± 122.2	18.01	1.03	<0.001	−4.73 to 234.93
Low-intensity running during the 4 × 4 min (m)	1020.4 ± 199.0 *	880.5 ± 160.9	6.10	0.76	<0.001	−54.16 to 333.96
Moderate-intensity running during the 4 × 4 min (m)	350.3 ± 87.4 *	285.8 ± 103.7	4.44	0.67	<0.01	−38.34 to 167.34
High-intensity running during the 4 × 4 min (m)	63.1 ± 31.3 *	47.7 ± 32.7	3.42	0.48	<0.05	−45.92 to 22.72
**Indicator of Workload**						
Player load (AU)	98.07 ± 12.5 *	89.4 ± 10.5	2.80	0.75	<0.05	−3.73 to 21.03
Maximal velocity (km/h)	20.8 ± 2.2	17.5 ± 1.8	1.79	1.64	<0.05	1.14 to 5.46
Power (w/kg)	8.10 ± 3.5	6.42 ± 2.9	1.84	0.52	<0.05	−1.77 to 5.13
**Sprints (>18 km·h^−1^)**						
Sprints duration (s)	1.7 ± 0.2	1.2 ± 0.3	1.62	1.96	<0.05	0.23 to 0.77
Sprints distance (m)	7.9 ± 2.3 *	5.2 ± 2.0	3.21	1.25	<0.05	0.39 to 5.01
**Acceleration and Deceleration Number (>3 m·s^−^²)**						
Acceleration	15.6 ± 2.75 *	12.5 ± 1.75	3.22	1.34	<0.05	0.63 to 5.57
Deceleration	17.3 ± 3.20 *	14.4 ± 2.55	5.49	1.0	<0.05	−0.20 to 6.00

*: Significant difference compared to SSG-SG. bpm, beats per minute; CI, confidence interval; d, Cohen’s test; t, Student *t*-test; AU, Arbitrary unit.

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
