# Peer review of "Physical and Physiological Responses during the Stop-Ball Rule During Small-Sided Games in Soccer Players"

_sports, 2019, doi:10.3390/sports7050117_

Reviewer 1 Report

I want to make two suggestions, in case they were of interest to the authors.

I know there is a lot of difficulty, but the sample is very small (only 4 repetitions of a task). It is suggested that the measure be taken more times, on different days.

The GPS was switched on once the warm-up was over. To avoid wasting time and intervene in mid-training with the players, it is suggested: turn on the GPS at the beginning of the session and then treat the resulting graph by cutting only task times.

Regards

Author Response

Reviewer 1:

We would like to thank so much the reviewers for their comments that have been so helpful in improving the manuscript’s quality.

Comments to the Author

I know there is a lot of difficulty, but the sample is very small (only 4 repetitions of a task). It is suggested that the measure be taken more times, on different days.

Players train daily with both types of SSG during training session. Thus, there is many repetition of the task.

The GPS was switched on once the warm-up was over. To avoid wasting time and intervene in mid-training with the players, it is suggested: turn on the GPS at the beginning of the session and then treat the resulting graph by cutting only task times.

The GPS device was activated 15 min before kick-off, in accordance with the manufacturer’s instructions.

Reviewer 2 Report

General Comments: The authors have submitted an interesting paper on different same-sided game strategies to examine the physical and physiological differences. While the manuscript has its strong points, there are a number of things that must be improved. First, the authors must stay consistent with the writing tense they are use. On several occasions, the authors switched from past tense to present or future tense. Second, the authors compared 14 different variables without correcting for error.  While this may not affect the practical results, it is important that the authors provide accurate statistical findings as well. Third, the authors should revisit the way that the data is presented in the Results section. Much of the section could be summarized in one table rather than three. Finally, the discussion and conclusions sections should be re-written as the former appears to be disjointed with incomplete thoughts and interpretations, while the latter needs to provides more thorough practical applications.

Specific Comments:

Line 26: Change 'was' to 'were' and modify the sentence to indicate that multiple t-tests were used.

Line 27: ...compared to SSG-SG

Line 32: Change to '...duration was found.'

Lines 48-55: While SSG-SG and SSG-SB are introduced here, it would be nice to provide the reader with a brief explanation of what each of these consist of.

Line 55 should be a continuation of the previous paragraph instead of a new one.  The new paragraph should start with the introduction of GPS.  Further information on this will be needed to expand on this.

Line 65: The word 'sought' may not be the most appropriate term here.

Line 73: Should 'young' be 'youth' instead to be consistent with the terminology used in the literature?

Lines 86-87: This should be introduced in the introduction to familiarize the readers with the terminology.

Line 90: This sentence is an example where the writing tense changes throughout the manuscript. Each aspect of the manuscript should be written in past tense.

Line 99: Has this product been validated within a study?  If so, the authors should cite that study here.

Lines 114-115: Check the writing tense here.

Line 121: Specify what variables were compared here.  In addition, it appears that the authors are comparing 14 different variables without a correction of family-wise Type I error.  It is recommended that the authors use a Bonferroni or Holm-Bonferroni correction.  Assuming the authors focused on the practical significance within the discussion, this should not require much modification.  However, adjusted p-values should be presented.  Finally, how was reliability assessed in the current study?

Lines 125-141 and Table 3: The authors already specified how the Cohen's d values should be interpreted in the section above. Thus, it is unnecessary to include the interpretation in the Results section as well.  I would also add that the entire results section could be summarized in one table with the means, standard deviations, effect sizes, and p-values.  The t-values may not need to be reported.  By summarizing the data in this manner, the discussion can focus on the magnitude of the differences rather than re-summarizing what the Results already say.

Discussion: Where there are aspects of the discussion that are well done, its current form is disjointed and difficult to read. The authors should re-write the discussion by merging common thoughts and ideas to a greater extent and trying to avoid short paragraphs of 2 or 3 sentences. This should allow the authors to explain their results and relate it to the relevant literature to a greater extent.

Conclusions: The conclusions should summarize the findings of the current study and then provide practical recommendations for coaches and practitioners. In its current form, it is lacking in this regard.  The main question the authors should attempt to answer for practitioners is "How can I as a practitioner use this information?"

Author Response

Reviewer: 2

General comments

The authors have submitted an interesting paper on different same-sided game strategies to examine the physical and physiological differences. While the manuscript has its strong points, there are a number of things that must be improved.

The authors must stay consistent with the writing tense they are use. On several occasions, the authors switched from past tense to present or future tense.

The manuscript has been revised in relation with the writing tense

The authors should revisit the way that the data is presented in the Results section. Much of the section could be summarized in one table rather than three.

Correction made as suggested. Please see changes made in the result section.

The discussion and conclusions sections should be re-written as the former appears to be disjointed with incomplete thoughts and interpretations, while the latter needs to provides more thorough practical applications.

Paragraphs were added to the discussion and the conclusion sections. Please see changes made in the text.

Specific Comments:

Line 26: Change 'was' to 'were' and modify the sentence to indicate that multiple t-tests were used.

Correction made as suggested. Please see changes made in the text.

Line 27: ...compared to SSG-SG

Correction made as suggested. Please see changes made in the text.

Line 32: Change to '...duration was found.'

Correction made as suggested. Please see changes made in the text.

Lines 48-55: While SSG-SG and SSG-SB are introduced here, it would be nice to provide the reader with a brief explanation of what each of these consist of.

Hope it’s clear. Please see changes made in the text.

Line 55 should be a continuation of the previous paragraph instead of a new one.  The new paragraph should start with the introduction of GPS.  Further information on this will be needed to expand on this.

Correction made as suggested. Please see changes made in the text.

Line 65: The word 'sought' may not be the most appropriate term here.

Correction made as suggested. Please see changes made in the text.

Line 73: Should 'young' be 'youth' instead to be consistent with the terminology used in the literature?

I think that "Young" is the appropriate sentence in our study, because "Youth" can  mean kid or teenager (usually around 13-17), and "Young" usually means someone whose between the ages of 17 and 25.

Lines 86-87: This should be introduced in the introduction to familiarize the readers with the terminology.

Correction made as suggested. Please see changes made in the text.

Line 90: This sentence is an example where the writing tense changes throughout the manuscript. Each aspect of the manuscript should be written in past tense.

Correction made as suggested. Please see changes made in the text.

Line 99: Has this product been validated within a study?  If so, the authors should cite that study here.

Similarly to previous study (Giménez et al., 2017 and Castellano et al., 2013) we have used the GPS tracker devices (Playertek, Catapult Innovations, sampling frequency = 10 Hz, Melbourne, Australia).

Lines 114-115: Check the writing tense here.

Correction made as suggested. Please see changes made in the text.

Line 121: Specify what variables were compared here.  In addition, it appears that the authors are comparing 14 different variables without a correction of family-wise Type I error.  It is recommended that the authors use a Bonferroni or Holm-Bonferroni correction.  Assuming the authors focused on the practical significance within the discussion, this should not require much modification.  However, adjusted p-values should be presented.  Finally, how was reliability assessed in the current study?

Concerning the use of Bonferroni correction, to our knowledge, there is no correction of t-test (two measures for one group) with Bonferroni. If there is a correction for the t-test please send us how we could calculated it.

Moreover, concerning reliability, players were familiarized previously with the two  types of SSG and the devices that would be used  during  the  study.  Intraclass  correlation  and the  coefficient  of variation (CV) were calculated in order to determine the reliability of the measured parameters. This has been added to the text (statistical analyses).

Lines 125-141 and Table 3: The authors already specified how the Cohen's d values should be interpreted in the section above. Thus, it is unnecessary to include the interpretation in the Results section as well.  I would also add that the entire results section could be summarized in one table with the means, standard deviations, effect sizes, and p-values.  The t-values may not need to be reported.  By summarizing the data in this manner, the discussion can focus on the magnitude of the differences rather than re-summarizing what the Results already say.

Correction made as suggested. Please see changes made in the text.

Discussion: Where there are aspects of the discussion that are well done, its current form is disjointed and difficult to read. The authors should re-write the discussion by merging common thoughts and ideas to a greater extent and trying to avoid short paragraphs of 2 or 3 sentences. This should allow the authors to explain their results and relate it to the relevant literature to a greater extent.

Paragraphs were added to the discussion section. Please see changes made in the text.

Conclusions: The conclusions should summarize the findings of the current study and then provide practical recommendations for coaches and practitioners. In its current form, it is lacking in this regard.  The main question the authors should attempt to answer for practitioners is "How can I as a practitioner use this information?"

Paragraph was added to the conclusion section. Please see changes made in the text.

Reviewer 3 Report

Brief summary

The study aimed to add another parameter in soccer small-sided games that allow manipulation of exercise intensity. Specifically, the parameter of interest is the method of scoring using “stopping the ball” rather than placing the ball in a goal. This is an emerging topic in small-sided games and the manuscript has potential to contribute to the understanding of effects of the parameter manipulation.

Broad comments

Strengths

1.       The study deals with an emerging topic that has not been examined extensively. Due to the lack of literature, the study makes contribution to understanding effects of manipulating the scoring method in small-sided games.

2.       The study quantified performance variables in addition to heart rate responses. Because heart rate can change in response to many factors outside physical activities, quantification of performance during small-sided games with such manipulation can help ensure that the manipulation of scoring method is in fact leading to greater exercise intensity.

3.       While the manuscript is not clear whether the authors used the (randomized) cross-over design, if they did, the sample size of 16 players should provide adequate statistical power.

Weaknesses

1.       It would be beneficial if the authors explain why understanding performance during small sided-games of different formats is important in the introduction.

2.       The study was conducted using players of the same team, presumably under the instruction of the same coaching staff. While this point should not prevent the paper from getting published, the authors should acknowledge that the observations made in the study could be at least partially due to the environment unique to the team and the coaching staff.

3.       The manuscript is not clear on the sex of the subjects – male, female, or mix. If a mix, what was the male and female breakdown?

4.       The manuscript may be improved if it provides more description of how to implement the stop the ball scoring system. Currently, for readers who are not familiar, all they can gather is that a point is scored when the ball is stopped inside a designated scoring zone. Possible questions that may help unfamiliar readers are “does the ball have to stop in the zone or all it takes is for the ball to enter the zone?”, “do you have to dribble the ball into the zone and stop or can a player pass the ball to another who then can stop the ball in the zone?”, and “is there a restriction on how long a player can stay in the zone?”.

5.       As noted above, the manuscript is not clear if 16 players were divided into two groups, one of which underwent the testing order of SB and then SG while the other underwent the order of SG and then SB. Or were the data collected in a completely different procedure? The key points that matter here as it relates to the statistical analysis and inferences are 1) did every player participate in both SB and SG? And 2) if so, was the order of participation (i.e. SB to SG vs. SG to SB) randomized for each player?

6.       It is not until the end of the manuscript that the authors disclose how max HR was calculated. This should appear in the methods.

7.       Addition of 95% confidence intervals for mean difference should further improve inferences that can be made from the study. Currently, all the inferences made in the manuscript are based on the outcome of null hypothesis testing (i.e. p values). Null hypothesis testing is certainly a convenient method of statistical inference. However, it is not perfect. Thus, inferences about the population that the study’s sample represents should be carried out more carefully while examining effect size and confidence intervals.
For example, the authors infer that “the findings of this study suggest that SSG-SB played with 4 vs. 4 format on constant pitch, provides a greater HR and physical responses than SSG-SG and its use in training elite soccer players is highly recommended”. While I see the possibility of this new manipulation to small-sided games, I believe it is too early to make such statement as “greater HR and physical responses” and “highly recommended”. The following are some of the major reasons why I believe it’s too early.
1) This study was cross-sectional in nature. Thus, while possibly higher responses were observed with the SB format, we do not know yet whether the SB format actually does induce favorable adaptations compared to the SG format. Only training studies can provide direct evidence.
2) Assuming that the all 16 players participated in both formats, the 95% confidence interval for the mean difference between the two formats for HR is 1.22 to 12.78 bpm. This means that in the population represented by the study’s sample, the average difference between the two formants in HR can be as small as 1.22 bpm. While certainly the average difference can be as large as 12.78 bpm, the magnitude of effect on HR does not appear clear enough to infer that training adaptations would be more favorable with the SB format in the population.
3)  The concern about the inference of the effect based on HR also applies to all the other variables, particularly the performance variables obtained from the GPS due to the reported minimal effect. Despite the rejection of the null hypothesis for all of them, the effect observed in the sample was considered minimal. This leaves a possibility that the average difference in the population is practically negligible or no effect, meaning that the ultimate training adaptations from the SB format may not be practically different from the SG format. For example, the 95% confidence interval for high-intensity running distance is 5.80 to 25.00m. Again, while the average difference between the two formats in the population can be as much as 25.00m, it can also be as little as 5.80m. The study provides no further evidence to favor either direction.
In summary of this point #6, I am not arguing that SB does not offer potential benefits over SG. What I am arguing is that the inferences seen in some statements made by the authors appear beyond the evidence provided by the study. Going back to the cited statement about SB providing greater responses and thus being highly recommended, I have to argue that this is an overstatement. The observations with the sample showed greater responses. The available statistical evidence suggests the same possibility in the population. When combined with other studies dealing with the same topic, the possibility may favor greater responses more. But the parameter estimates (e.g. confidence intervals) are not very clear and the lack of evidence from training studies makes it difficult to highly recommend SB over SG. At this point, what we are sure about is that SB appears at least as an alternative to SG and holds a possibility of greater adaptations for soccer players compared to SG.

Specific comments

Line 20 – “… have not investigated” should be “have not been investigated”

Line 37 – “Soccer is a highly physical demanding sport, …” should be “Soccer is a highly physically demanding sport…”

Line 40 – “upmost” should be “utmost”

Line 44 – “its widely utilization” should be “its wide utilization”

Line 49 – I think “… induced higher intensities 9i.e., higher HR values) than small-goal SSG…” would read better if it’s “…induced higher HR values than small-goal SSG…”

Line 57 – “… have been reported as to …” would make sense if it’s “… have been reported for …”

Line 62 – “… no previous study has comparing the physical response…” should be “… no previous study has compared the physical response…”

Line 65 – “We sought that conclusions …” makes more sense if the authors delete the “We sought that”.

Line 78 – “the same number of player” should be “the same number of players”

Line 81 – “…, they realized a supplementary …” would make more sense if it’s “…, they underwent a supplementary …”

Line 84 – “… and SSG-SG has been realized in different sessions …” would make more sense if it’s “… and SSG-SG were conducted on different days …”

Line 85 – I believe the authors meant “A point was scored …” by “A point has been obtained …”

Line 86 – “playing surface” sounds better with “playing field”.

Line 91 – “To minimize the lost time, balls were …” would make sense if it’s “To minimize the lost time, extra balls were …”

Line 98 – the “e.g.” in “Data (e.g., acceleration, …” implies that there were other types of measurement beside what is listed here or did the authors meant “i.e.”?

Line 102-104 – “After recording, …” is an incomplete sentence.

Line 104 – Provide justifications for using these velocity bands.

Line 111 – the equation for player load here does not make sense. Each of the 3 pairs of parentheses basically takes the form of acceleration x time = i + 1 – acceleration x time = 1. This basically reduces to acceleration x time = 1. Thus, the equation will always reduce to player load = square root of 1/100.

Line 114 – what is the difference between current time (i) and time (t)?

Line 114 – “… has been …” should be “was”.

Line 160 – did the authors mean “… less technical demands during …” by “…less technical abilities during…”?

Line 177 – “… between two games rules” should be “… between two game rules”

Line 183 – “… in play during longer time…” should be “in play for longer time”

Line 196 – “… official matches demands” should be “official matches” – no demands

Line 197 – does it make more sense if “… to allow faster recovery” is “…to prevent unnecessary fatigue”?

Author Response

Reviewer: 3

Strengths

1.       The study deals with an emerging topic that has not been examined extensively. Due to the lack of literature, the study makes contribution to understanding effects of manipulating the scoring method in small-sided games.

2.       The study quantified performance variables in addition to heart rate responses. Because heart rate can change in response to many factors outside physical activities, quantification of performance during small-sided games with such manipulation can help ensure that the manipulation of scoring method is in fact leading to greater exercise intensity.

3.       While the manuscript is not clear whether the authors used the (randomized) cross-over design, if they did, the sample size of 16 players should provide adequate statistical power.

Weaknesses

It would be beneficial if the authors explain why understanding performance during small sided-games of different formats is important in the introduction.

I think that including SSG performance in the introduction is necessary because SSG training modality promotes significant performance  enhancement  and  training eciency. This method  of  training  satisfy  its  application within elite level soccer as a conditioning stimulus capable of improving aerobic endurance capacity.Hope’s its clear now.

The study was conducted using players of the same team, presumably under the instruction of the same coaching staff. While this point should not prevent the paper from getting published, the authors should acknowledge that the observations made in the study could be at least partially due to the environment unique to the team and the coaching staff.

Paragraph was added to the limitation part. Please see changes made in the text.

The manuscript is not clear on the sex of the subjects – male, female, or mix. If a mix, what was the male and female breakdown?

In this manuscript, young male soccer players participated. Please see changes made in the text.

The manuscript may be improved if it provides more description of how to implement the stop the ball scoring system. Currently, for readers who are not familiar, all they can gather is that a point is scored when the ball is stopped inside a designated scoring zone. Possible questions that may help unfamiliar readers are “does the ball have to stop in the zone or all it takes is for the ball to enter the zone?”, “do you have to dribble the ball into the zone and stop or can a player pass the ball to another who then can stop the ball in the zone?”, and “is there a restriction on how long a player can stay in the zone?”.

Paragraph was added. Hope it’s clear now.

As noted above, the manuscript is not clear if 16 players were divided into two groups, one of which underwent the testing order of SB and then SG while the other underwent the order of SG and then SB. Or were the data collected in a completely different procedure? The key points that matter here as it relates to the statistical analysis and inferences are 1) did every player participate in both SB and SG? And 2) if so, was the order of participation (i.e. SB to SG vs. SG to SB) randomized for each player?

In a randomized counterbalanced order, the players performed the training sessions: SSG-SB and SSG-SG. Hope it’s clear now.     

It is not until the end of the manuscript that the authors disclose how max HR was calculated. This should appear in the methods.

Correction made as suggested. Please see changes made in the text.

Addition of 95% confidence intervals for mean difference should further improve inferences that can be made from the study. Currently, all the inferences made in the manuscript are based on the outcome of null hypothesis testing (i.e. p values). Null hypothesis testing is certainly a convenient method of statistical inference. However, it is not perfect. Thus, inferences about the population that the study’s sample represents should be carried out more carefully while examining effect size and confidence intervals.

Correction made as suggested. Please see changes made in the table.
For example, the authors infer that “the findings of this study suggest that SSG-SB played with 4 vs. 4 format on constant pitch, provides a greater HR and physical responses than SSG-SG and its use in training elite soccer players is highly recommended”. While I see the possibility of this new manipulation to small-sided games, I believe it is too early to make such statement as “greater HR and physical responses” and “highly recommended”. The following are some of the major reasons why I believe it’s too early. 

Correction made as suggested. Please see changes made in the text.
1) This study was cross-sectional in nature. Thus, while possibly higher responses were observed with the SB format, we do not know yet whether the SB format actually does induce favorable adaptations compared to the SG format. Only training studies can provide direct evidence.

Your comment is very relevant. In further studies, we can explore the physical and the physiological adaptations of SSG-SB and SSG-SG with training studies.
2) Assuming that the all 16 players participated in both formats, the 95% confidence interval for the mean difference between the two formats for HR is 1.22 to 12.78 bpm. This means that in the population represented by the study’s sample, the average difference between the two formants in HR can be as small as 1.22 bpm. While certainly the average difference can be as large as 12.78 bpm, the magnitude of effect on HR does not appear clear enough to infer that training adaptations would be more favorable with the SB format in the population. The concern about the inference of the effect based on HR also applies to all the other variables, particularly the performance variables obtained from the GPS due to the reported minimal effect. Despite the rejection of the null hypothesis for all of them, the effect observed in the sample was considered minimal. This leaves a possibility that the average difference in the population is practically negligible or no effect, meaning that the ultimate training adaptations from the SB format may not be practically different from the SG format. For example, the 95% confidence interval for high-intensity running distance is 5.80 to 25.00m. Again, while the average difference between the two formats in the population can be as much as 25.00m, it can also be as little as 5.80m. The study provides no further evidence to favor either direction.

We totally agree with your comment. However, we didn’t see in our manuscript the value of 95% confidence interval of Heart rate that you reported. In our study and in order to establish whether the average difference between the training adaptations of the two SSG format, we based our study of Cohen’s  effect  size  criterion.  Quantitative  chances  of  higher  or  lower differences were evaluated qualitatively as follows no effect (d<0.41), minimum (0.41<d<1.15), moderate (1.152.70). In addition, we added in the present version the IC95% of each effect size in order to have a better idea about the average difference between the two SSG format.
In summary of this point #6, I am not arguing that SB does not offer potential benefits over SG. What I am arguing is that the inferences seen in some statements made by the authors appear beyond the evidence provided by the study. Going back to the cited statement about SB providing greater responses and thus being highly recommended, I have to argue that this is an overstatement. The observations with the sample showed greater responses. The available statistical evidence suggests the same possibility in the population. When combined with other studies dealing with the same topic, the possibility may favor greater responses more. But the parameter estimates (e.g. confidence intervals) are not very clear and the lack of evidence from training studies makes it difficult to highly recommend SB over SG. At this point, what we are sure about is that SB appears at least as an alternative to SG and holds a possibility of greater adaptations for soccer players compared to SG.

Correction made as suggested. Please see changes made in the table.

Specific comments

Line 20 – “… have not investigated” should be “have not been investigated”

Correction made as suggested. Please see changes made in the text.

Line 37 – “Soccer is a highly physical demanding sport, …” should be “Soccer is a highly physically demanding sport…”

Correction made as suggested. Please see changes made in the text.

Line 40 – “upmost” should be “utmost”

Correction made as suggested. Please see changes made in the text.

Line 44 – “its widely utilization” should be “its wide utilization”

Correction made as suggested. Please see changes made in the text.

Line 49 – I think “… induced higher intensities 9i.e., higher HR values) than small-goal SSG…” would read better if it’s “…induced higher HR values than small-goal SSG…”

Correction made as suggested. Please see changes made in the text.

Line 57 – “… have been reported as to …” would make sense if it’s “… have been reported for …”

Correction made as suggested. Please see changes made in the text.

Line 62 – “… no previous study has comparing the physical response…” should be “… no previous study has compared the physical response…”

Correction made as suggested. Please see changes made in the text.

Line 65 – “We sought that conclusions …” makes more sense if the authors delete the “We sought that”.

Correction made as suggested. Please see changes made in the text.

Line 78 – “the same number of player” should be “the same number of players”

Correction made as suggested. Please see changes made in the text.

Line 81 – “…, they realized a supplementary …” would make more sense if it’s “…, they underwent a supplementary …”

Correction made as suggested. Please see changes made in the text.

Line 84 – “… and SSG-SG has been realized in different sessions …” would make more sense if it’s “… and SSG-SG were conducted on different days …”

Correction made as suggested. Please see changes made in the text.

Line 85 – I believe the authors meant “A point was scored …” by “A point has been obtained …”

Correction made as suggested. Please see changes made in the text.

Line 86 – “playing surface” sounds better with “playing field”.

Correction made as suggested. Please see changes made in the text.

Line 91 – “To minimize the lost time, balls were …” would make sense if it’s “To minimize the lost time, extra balls were …”

Correction made as suggested. Please see changes made in the text.

Line 98 – the “e.g.” in “Data (e.g., acceleration, …” implies that there were other types of measurement beside what is listed here or did the authors meant “i.e.”?

(i.e) is the correct abbreviation. Please see changes made in the text.

Line 102-104 – “After recording, …” is an incomplete sentence.

Sentence is completed. Please see changes made in the text.

Line 104 – Provide justifications for using these velocity bands.

Similarly to previous study (Casamishana et Castellano, 2010 ; Hill-Hass et al., 2009 ; Rebelo et al., 2016) we have used these velocity bands.

Line 111 – the equation for player load here does not make sense. Each of the 3 pairs of parentheses basically takes the form of acceleration x time = i + 1 – acceleration x time = 1. This basically reduces to acceleration x time = 1. Thus, the equation will always reduce to player load = square root of 1/100.

I suggest to delete this pragraph.

Line 114 – what is the difference between current time (i) and time (t)?

Suggestion to delete this paragraph

Line 114 – “… has been …” should be “was”.

Correction made as suggested. Please see changes made in the text.

Line 160 – did the authors mean “… less technical demands during …” by “…less technical abilities during…”?

I think that « less technical abilities » is the appropriate sentence

Line 177 – “… between two games rules” should be “… between two game rules”

Correction made as suggested. Please see changes made in the text.

Line 183 – “… in play during longer time…” should be “in play for longer time”

Correction made as suggested. Please see changes made in the text.

Line 196 – “… official matches demands” should be “official matches” – no demands

Correction made as suggested. Please see changes made in the text.

Line 197 – does it make more sense if “… to allow faster recovery” is “…to prevent unnecessary fatigue”?

Correction made as suggested. Please see changes made in the text.

Round  2

Reviewer 2 Report

The authors have done well to address all of my major concerns.

This manuscript is a resubmission of an earlier submission. The following is a list of the peer review reports and author responses from that submission.

Round  1

Reviewer 1 Report

I believe that some aspects of the Experimental Procedures section should be improved. 

These are the following:

- Authors must indicate the number of times that each task was measured 4vs.4 (SSG-SB and SSG-SB.) If the task was only measured once, the "n" used has little reliability.

- I suggest that briefly, indicate the contents of the warm-up

- Authors should indicate why they used 2-3m.s2 and 3 m.2 for acceleration and deceleration. Two bibliographic references included by the authors, if they correspond to the speed ranges used in the study. However, these cited studies do not speak either of the range of acceleration and deceleration. Authors are asked to quote an article that provides identical or similar ranges.

- The authors should explain better how they obtained the HRmax. This concept has little consistency without a previous test. If this HRmax was obtained during the task, it may not be the real HRmax of each player. Perhaps the task does not cause the HRmax of the player to be reached.

I hope my English is understood by the editor and the authors.

I remain available to those interested.

A greeting

Author Response

Reviewer 1:

We would like to thank so much the reviewer for their comments that have been so helpful in improving the manuscript’s quality.

Comments to the Author

Authors must indicate the number of times that each task was measured 4vs.4 (SSG-SB and SSG-SB.) If the task was only measured once, the "n" used has little reliability.

A paragraph was added to the experimental approach. Please see changes made in the revised version.

I suggest that briefly, indicate the contents of the warm-up

    Correction made as suggested. Please see changes made in the revised version.

Authors should indicate why they used 2-3m.s2 and      3 m.2 for acceleration and deceleration. Two bibliographic references      included by the authors, if they correspond to the speed ranges used in      the study. However, these cited studies do not speak either of the range      of acceleration and deceleration. Authors are asked to quote an article      that provides identical or similar ranges.

Reference was added. Please see changes made in the revised version

The authors should explain better how they      obtained the HRmax. This concept has little consistency without a previous      test. If this HRmax was obtained during the task, it may not be the real      HRmax of each player. Perhaps the task does not cause the HRmax of the      player to be reached.

We calculated HRmax with this equation: HRmax: 220- age.  

% HRmax : (((HRmax – HR) / HRmax)) ×100)- 100

Reviewer 2 Report

in conclusion section provide a concise conclusion acording to the aims of the reseach. Idem in this section of the abstract

Author Response

Reviewer: 2

Comments to the Author

In conclusion section provide a concise conclusion according to the aims of the research. Idem in this section of the abstract.

Correction made as suggested. Hope it’s clear. Please see changes made in the revised Version

Reviewer 3 Report

Abstract needs to be improved. Insert the statistical method in the abstract. Put the values for “higher” on the physical variables (L28-29).

The introduction needs some work. I do not think it provides a succinct statement or background of the study, only that “SSG-SB is not investigated yet”. The justification and practical importance of the study should be clearly stated. I would appreciated a more thorough argument for the need and importance of this study.

Why only compare SSG-SB with SSG small goals?

Method.

The method section needs to be of higher quality. I don’t think a colleague can reproduce the experiment and get the same outcome. This is mainly because:

·       It is not described the number of matches for the different conditions that were included in the analysis.

·       No definition of Player Load values or indicators of workload.

·       No definitions of sprints in the method section but sprints is a variable in Table 3.

·       How long do the players need to accelerate/decelerate in order to count as an acceleration? How long do the player stay over a speed limit in order to count in the different categories?

·       Specify what Student’ t-test (paired?). Must evaluate the same player in different conditions. What is done with the missing?

·       Reliability of the tracking system for the different variables?

·       It is described that “the player were instructed to move and not to defend the  goal by staying nearby the goal…”. Did they? What if they were one goal up? This is of great importance for the intensity of the games.  

Results

The result section in the present form not of quality for a publication in Sports. This is mainly because:

·       All tables needs to be changed. There is no description of the time interval the data represents in the tables.

·       Table 3: Maximal velocity must have wrong SI unit, I don’t think >20 m.s-1 is possible for humans since that is approximately 70 km.h-1.

·       Check also units for sprints and acc/dec in table 3

·       No description of Player Load, Power or Maximal velocity or Sprints in the method.

·       Since it is a low number of  players you should consider showing individual data for the two conditions. Perhaps Modified Brinley plots or something similar…

Discussion

·       Need to see the discussion again after changes have been made to the method and the results.

·       Some other comments:

o   P5- Line 138, Is heart rate a good method to regulate and evaluate the intensity in SSG?

o   The HR-intensity during SSG-SB is not higher than many investigation of 4vs4 with goalkeeper, is it perhaps only the small goals that leads to decrease in intensity? This needs to be discussed.

o   P5-Line147: if it is a new form for scoring, perhaps there is less tactics and more running?

o   P5-Line162.. : Distance at high speed per minute in SSG is of other found to be less than the average for high speed running per minute in a match (see for example: Dalen, T., Sandmæl, S., Stevens, T. G. A., Hjelde, G. H., Kjøsnes, T. N., & Wisløff, U. (2019). Differences between acceleration and high intensity activities in small-sided games and peak periods of official matches in elite soccer players. Journal of Strength & Conditioning Research. How is it in this study? If it’s the same, is it a good measure of intensity during SSG?

o   P6-Line 195-197. Consider reformulate. I don’t think this study is  of great practical relevance… but it might give additional information.

Author Response

Reviewer: 3

Comments to the Author

Abstract needs to be improved. Insert the statistical method in the abstract. Put the values for “higher” on the physical variables (L28-29).

Statistical method was added. Please see changes made in the text.

Values were added. Please see changes made in the text.

The introduction needs some work. I do not think it provides a succinct statement or background of the study, only that “SSG-SB is not investigated yet”. The justification and practical importance of the study should be clearly stated. I would appreciated a more thorough argument for the need and importance of this study.

Correction made as suggested. Hope it’s clear. Please see changes made in the revised version.

Why only compare SSG-SB with SSG small goals?

In previous studies (Halouani et al., 2014, 2017a and 2017b), these 2 games rules are studied only with physiological responses. In our study we have interest in the physical responses to confirm the effectiveness of these 2 game rules in improving training intensity (physical and physiological responses).

Method.

The method section needs to be of higher quality. I don’t think a colleague can reproduce the experiment and get the same outcome. This is mainly because:

It is not described the number of matches for the different conditions that were included in the analysis.

Description was added. Please see changes made in the revised version.

No definition of Player Load values or indicators of workload.

 Corrected. Please see changes made in the revised version

No definitions of sprints in the method section but sprints is a variable in Table 3.

Correction made as suggested. Hope it’s clear. Please see changes made in the revised version.

How long do the players need to accelerate/decelerate in order to count as an acceleration? How long do the player stay over a speed limit in order to count in the different categories?

The players need to accelerate/decelerate one time to count as an acceleration or deceleration. Concerning speed, each interval of speed was calculated with GPS according to the time and distance that the player maintain the required intensity.

Specify what Student’ t-test (paired?). Must evaluate the same player in different conditions. What is done with the missing?

In this study, we have used paired t-test of student and all players realized the two conditions. Please see changes made in the revised version

Reliability of the tracking system for the different variables?

According to Hill-Hass et al. (2009), the reliability of the GPS has reported with the typical error expressed as a coefficient of variation being 3.6% for total distance, 4.3% for low-intensity activity (0–6.9 km.h -1 ), 11.2% for higher intensity running (>14.4 km.h -1 ), and 5.8% for peak speed.

It is described that “the player were instructed to move and not to defend the  goal by staying nearby the goal…”. Did they? What if they were one goal up? This is of great importance for the intensity of the games.  

Yes, all participants were asked to defend and not allowed to stay nearby the goal (i.e., 2 m). Moreover, there are goals that are scored, these goals can enhance players’ motivation and the intensity of SSG.

Results

The result section in the present form not of quality for a publication in Sports. This is mainly because:

 All tables needs to be changed. There is no description of the time interval the data represents in the tables.

The data was performed according to the duration of SSG and was presented in the table in term of distance.

Table 3: Maximal velocity must have wrong SI unit, I don’t think >20 m.s-1 is possible for humans since that is approximately 70 km.h-1.

Correction of the unit was made. Please see changes in table 3.

Check also units for sprints and acc/dec in table 3

I think that the unit of sprints is mentioned in table 3 (Sprints duration « s » ; Sprints distance « m »). Concerning acc/dec, we have measured the number (Acceleration and deceleration number ; >3 m.s-²).

No description of Player Load, Power or Maximal velocity or Sprints in the method.

Description is added. Please see changes made in the revised version.

Since it is a low number of  players you should consider showing individual data for the two conditions. Perhaps Modified Brinley plots or something similar…

The reviewer ios right; however, when we perform the spaghetti graph, the data will not be clear; Thus, we suggest that we keep the table for the presentation of the data.

Discussion

Need to see the discussion again after changes have been made to the method and the results.

There’s no change in the results. So, we keep the same discussion.

·       Some other comments:

   P5- Line 138, Is heart rate a good method to regulate and evaluate the intensity in SSG?

Yes, Heart rate response  is a useful method to regulate the intensity of training in Soccer SSG.

The HR-intensity during SSG-SB is not higher than many investigation of 4vs4 with goalkeeper, is it perhaps only the small goals that leads to decrease in intensity? This needs to be discussed.

A paragraph was added in this context. Please see changes made in the text.

P5-Line147: if it is a new form for scoring, perhaps there is less tactics and more running?

Thanks for the new information and it’s included on the manuscript. Please see changes made in the revised version

 P5-Line162.. : Distance at high speed per minute in SSG is of other found to be less than the average for high speed running per minute in a match (see for example: Dalen, T., Sandmæl, S., Stevens, T. G. A., Hjelde, G. H., Kjøsnes, T. N., & Wisløff, U. (2019). Differences between acceleration and high intensity activities in small-sided games and peak periods of official matches in elite soccer players. Journal of Strength & Conditioning Research. How is it in this study? If it’s the same, is it a good measure of intensity during SSG?

SSG is based of a reduced pitch dimension. Thus, speed can’t be reached compared to soccer match. Sprint distance in soccer match is higher than SSG, but SSG training can improve players’ speed.

P6-Line 195-197. Consider reformulate. I don’t think this study is  of great practical relevance… but it might give additional information.

Corrected. Please see changes in the text.

Round  2

Reviewer 3 Report

Comments to the Author

Method.

It is not described the number of matches for the different conditions that were included in the analysis.

Is there only one training session for each condition? “…SSG-SB and SSG-SG has been realized in different sessions (i.e., one for SSG-SB and one for SSG-SG) at the same time of day (16h00 to 18h00)”

If this is the case, I don’t see you have enough data to make proper statistical analysis (weather and other external factors could huge part of the reason for the difference between conditions). If its not the case, you have to inform me and the reader about the number of SSG sessions for each condition.

No definition of Player Load values or indicators of workload.

            Still no definition of these variables, only written “that they were recorded”.

No definitions of sprints in the method section but sprints is a variable in Table 3.

            Still no definition of the speed zone for Sprints?

Your % of HR-max values have no value if they are based on age (220-age). There is to high individual variability within a soccer team for this equation to have any value.

How long do the players need to accelerate/decelerate in order to count as an acceleration? How long do the player stay over a speed limit in order to count in the different categories?

So you don’t have to accelerate a specific time  (i.e. >0.5 sec or something) in order to count as an acceleration? Then 0,5m change of directions could be measured and counted as acceleration and deceleration? Is there a pr

Specify what Student’ t-test (paired?). Must evaluate the same player in different conditions. What is done with the missing?

I need more information about the statistical analysis in the article. As I assume (but still cannot read from the tables) Table 2 represent the total distance moved in 16 minutes of SSG-conditions. If this is the data that were analyzed you only have one session for each condition per player? More information is needed…           

Reliability of the tracking system for the different variables?

According to Hill-Hass et al. (2009), the reliability of the GPS has reported with the typical error expressed as a coefcient of variation being 3.6% for total distance, 4.3% for low-intensity activity (0–6.9 km.h -1 ), 11.2% for higher intensity running (>14.4 km.h -1 ), and 5.8% for peak speed.

These reliability values are according to Hill_Haas (2010), not 2009… I addition Hill Haas doesn’t use the same system, and different sampling frequency, so you can’t use their reliability data to address your system.

Results

All tables needs to be changed. There is no description of the time interval the data represents in the tables.

The data was performed according to the duration of SSG and was presented in the table in term of distance.

But why not put in the table that it is distance from 4*4minutes of SSG (or something like that). Then the reader know its distances from 16 minutes (if that’s correct).

Check also units for sprints and acc/dec in table 3

I think that the unit of sprints is mentioned in table 3 (Sprints duration « s » ; Sprints distance « m »). Concerning acc/dec, we have measured the number (Acceleration and deceleration number ; >3 m.s-²).

            Sprints should be (>18 km.h-1) not  (>18 km.h-1)

Acceleration and deceleration should be (>3 m.s-2) not  (>3 m.s-2)

No description of Player Load, Power or Maximal velocity or Sprints in the method.

Still no description, only written “that they were recorded”.

Author Response

Point-by-point response to the reviewers

         We thank the reviewer and the editor for their thorough review of our work and for the very constructive and helpful comments. We have taken the comments into consideration and have provided specific responses for each reviewer. Our responses appear in blue typeface. We hope that this version has been improved and that is now suitable for publication in your journal. Furthermore, we are ready to make any further changes that would be deemed necessary for any deeper improvement.
Reviewer(s)' Comments to Author:

Method.

It is not described the number of matches for the different conditions that were included in the analysis.

Is there only one training session for each condition? “…SSG-SB and SSG-SG has been realized in different sessions (i.e., one for SSG-SB and one for SSG-SG) at the same time of day (16h00 to 18h00)”

If this is the case, I don’t see you have enough data to make proper statistical analysis (weather and other external factors could huge part of the reason for the difference between conditions). If its not the case, you have to inform me and the reader about the number of SSG sessions for each condition.

The players performed for many times the two formats of the small sided games before the experimental phase. However, for the experimental phase all parameters were recorded for only one for SSG-SB and one for SSG-SG. The two conditions were performed in a random order.

No definition of Player Load values or indicators of workload.

            Still no definition of these variables, only written “that they were recorded”.

Correction was made. Hope’s it’s clear. Please see changes made in the revised version.

No definitions of sprints in the method section but sprints is a variable in Table 3.

            Still no definition of the speed zone for Sprints?

Correction was made. Hope’s it’s clear. Please see changes made in the revised version.

Your % of HR-max values have no value if they are based on age (220-age). There is to high individual variability within a soccer team for this equation to have any value.

The reviewer is right; However, as previous study have utilized this equation and for optimal comparison in the discussion we opted for this equation. We added this information in the limitation of the study.

How long do the players need to accelerate/decelerate in order to count as an acceleration? How long do the player stay over a speed limit in order to count in the different categories?

So you don’t have to accelerate a specific time  (i.e. >0.5 sec or something) in order to count. as an acceleration? Then 0,5m change of directions could be measured and counted as acceleration and deceleration? Is there a pr

The reviewer is right, the needed acceleration was counted by the GPS for >3 m·s-2.

Specify what Student’ t-test (paired?). Must evaluate the same player in different conditions. What is done with the missing?

Yes, all sixteen players have performed the two conditions. Thus, we utilized the Student’ t-test.

I need more information about the statistical analysis in the article. As I assume (but still cannot read from the tables) Table 2 represent the total distance moved in 16 minutes of SSG-conditions. If this is the data that were analyzed you only have one session for each condition per player? More information is needed…

 Yes the distance was calculated for the 16 min in each condition. In fact, all players have performed the two conditions. Thus, we utilized the Student’ t-test. 

Reliability of the tracking system for the different variables?

According to Hill-Hass et al. (2009), the reliability of the GPS has reported with the typical error expressed as a coefficient of variation being 3.6% for total distance, 4.3% for low-intensity activity (0–6.9 km.h -1 ), 11.2% for higher intensity running (>14.4 km.h -1 ), and 5.8% for peak speed.

These reliability values are according to Hill_Haas (2010), not 2009… I addition Hill Haas doesn’t use the same system, and different sampling frequency, so you can’t use their reliability data to address your system.

The reviewer is right; However, as recent researches have reported that this system is a valid and reliable marker of assessment for monitoring team player’s movement demands (Jennings et al., 2010 ; Mac Leod et al., 2009 ) we didn’t record the reliability of this devise in the players of the present study.

Results

All tables needs to be changed. There is no description of the time interval the data represents in the tables.

The data was performed according to the duration of SSG and was presented in the table in term of distance.

But why not put in the table that it is distance from 4*4minutes of SSG (or something like that). Then the reader know its distances from 16 minutes (if that’s correct).

Correction made as suggested ; please see changes made in the table.

Check also units for sprints and acc/dec in table 3

I think that the unit of sprints is mentioned in table 3 (Sprints duration « s » ; Sprints distance « m »). Concerning acc/dec, we have measured the number (Acceleration and deceleration number ; >3 m.s-²).

            Sprints should be (>18 km.h-1) not  (>18 km.h-1)

Acceleration and deceleration should be (>3 m.s-2) not  (>3 m.s-2)

 Correction made as suggested. Please see changes made in table 3.

No description of Player Load, Power or Maximal velocity or Sprints in the method.

Correction was made. Hope’s it’s clear. Please see changes made in the revised version.

Round  3

Reviewer 3 Report

The authors presents good knowledge on the topic, and the article well written. 

The main limitation is the design of the research. In my opinion it is a huge limitation that the data only includes one session for each condition. The variability between session is rather high based on different external conditions, and this variability would have great influence on the investigated variables. In addition it is difficult to see how you can randomize two sessions? 

I'm sorry, but I think the authors should collect more data, before they can evaluate differences between SSG-conditions.